# Aquatic invasive alien rodents in Western France: Where do we stand today after decades of control?

**Manon Bonnet[1,2], Gérald Guédon[3]\*, Marc Pondaven[3], Sandro Bertolino[4], Damien Padiolleau[5], Vanessa Pénisson[6], Francine Gastinel[7], Fabien Angot[8], Pierre-Cyril Renaud[1], Antonin Frémy[9], Olivier Pays[1,2]\***

**1** LETG-Angers, UMR 6554 CNRS, Université d'Angers, Angers, France, **2** REHABS International Research Laboratory, CNRS-Université Lyon 1-Nelson Mandela University, George, South Africa, **3** Polleniz, Beaucouzé, France, **4** Dipartimento di Scienze della Vita e Biologia dei Sistemi, Università degli Studi di Torino, Torino, Italy, **5** Polleniz 44, Grandchamp-des-Fontaines, France, **6** Polleniz 85, BP 141, La Roche Sur Yon Cedex, France, **7** Polleniz 53, Zone artisanale, Changé, France, **8** Polleniz 72, ZA de la Belle Croix, Requeil, France, **9** FDGDON 49, Beaucouzé, France

\* olivier.pays@univ-angers.fr (OP); Gerald.GUEDON@polleniz.fr (GG)

**Data Availability Statement:** All relevant data are within the manuscript and its Supporting Information files.

**Funding:** "Polleniz" provided support in the form of salaries for GG, MP, DP, VP, FG, FA and

## Abstract

Two aquatic invasive alien rodents, the coypu (*Myocastor coypus*) and muskrat (*Ondatra zibethicus*), have taken over a significant amount of wetlands in France. Pays de la Loire is an administrative region of about 32 000 km$^2$ in the Western France with 6.3% of its area in wetlands (excluding the Loire River). Populations of coypus and muskrats are established and a permanent control programme has been set to reduce their impacts. The control plan is based on few professional trappers and many volunteers which makes this programme unique compared to other programme relying on professionals only. The aim of this study is to analyse the temporal and spatial dynamics of coypu and muskrat captures during the last 10 years to evaluate their effectiveness. The number of rodents removed per year increased by 50% in 10 years and reached about 288 000 individuals in 2016 with about 80% of them being coypus. During the same time length, the number of trappers involved in the programme also increased by 50% to reach 3 000 people in 2016. Although the raise of coypus and muskrats trapped can possibly be explained by an increase of the number of trappers, the number of coypus removed per trapper per year increased by 22%. Despite the outstanding number of individuals removed per year, our results suggest that the programme does not limit the population dynamics of coypus. Finally, since 2017, the number of data gathered from municipalities decreased, as did the total number of individuals trapped. Indeed, although rewards are crucial to recruit new volunteers, subsidies from local and regional authorities are declining. Decision makers and financers should be encouraged to fund this programme from the perspectives of the direct or indirect costs related to the presence of aquatic invasive alien rodents in wetlands.

"FDGDON49" for AF. Each departmental federation for pest control ("Polleniz 44", "Polleniz 85", "Polleniz 72", "Polleniz 53", and "FDGDON49") has communicated all captures from trappers from 2008 to 2019. "Polleniz" and "FDGDON49" did not have any additional role in the study design, data collection and analysis, decision to publish, or preparation of the manuscript. The specific roles of these authors are articulated in the 'author contributions' section.

**Competing interests:** "Polleniz" provided support in the form of salaries for GG, MP, DP, VP, FG, FA, and "FDGDON49" for AF. This does not alter the adherence to PLOS ONE policies on sharing data and materials.

# Introduction

Invasive alien species (IAS) are considered as one of the main threats to native biodiversity [1–3]. They trigger important losses of native species and habitats and alter ecosystem functioning [4]. They can also carry infectious diseases threatening native species and human health [5]. Freshwater ecosystems are particularly impacted by invasive species [6] and the effective management of invasion is a priority. Even though global management frameworks exist and can be applied to tackle some biological invasions [7–10], the lack of solutions to limit their spatial expansion and associated costs makes management decisions against IAS very challenging [11].

According to the stage of the invasion process, a hierarchical strategy has been proposed to mitigate the negative impacts of invasive species, ranging from prevention of new introductions, eradication of newly established species, to spatial containment and/or population control programme [1, 12]. In several cases, eradication has been proved to be an effective conservation action, both on islands and on the mainland [13, 14]. However, when a species is widespread and the complete removal of individuals is not considered practical anymore, a permanent control programme promoting actions to mitigate its negative impacts could be considered as a possible alternative approach [15]. Though, control activities are very complex to implement in the field and maintain them for long-term as well can become very expensive [16–18].

In Europe, two semi-aquatic rodents, the coypu (*Myocastor coypus*) and muskrat (*Ondatra zibethicus*), have been included in the list of Invasive Alien Species of Union Concern (EU Regulation No 1143/2014). These two aquatic invasive alien rodents (AIAR) are widespread in Europe [19–21] and have been found across many wetlands in France [22, 23]. On top of this, they are important carriers of diseases contagious to humans, including leptospirosis [24], toxoplasmosis [25] and alveolar echinococcosis [26], and livestock and pets, including leptospirosis, giant liver fluke and salmonellosis [24, 27]. Both species dig burrows to shelter from climatic conditions and breed. The number of burrows is a significant siltation factor, and thus contribute to reduce drastically the water flow in some areas [28]. Moreover, burrows make riverbanks unstable and sometimes may collapse [28]. Although the real costs of AIAR on agricultural damages are largely unknown, several studies have reported that crops consumption close to streams and marshes may be high [16, 29]. Finally, it has been documented that coypus and muskrats may affect waterbirds in Italy [30], bivalve molluscs in USA [31] and France [32] and a large range of invertebrate species in Finland [33]. Studies have also proved that these species could impact native vegetation [34–37]. To sum up, it is commonly assumed that coypus and muskrats could negatively impact wetlands although more studies are needed to quantify their impacts on biodiversity and ecosystem services.

In France, despite the establishment of coypus' and muskrats' populations [24, 38], few studies reporting the current management strategies for these species and their results at the national or regional level have been published, particularly since the 1990s [39]. Indeed, there is no national data on the number of coypus and muskrats that are trapped each year although a study in 2016 indicated that around 350 000 coypus and 70 000 muskrats were shot during the 2013–2014 game season in France [40]. Unpublished data of captures exist but mainly at local levels as for instance in some municipalities, protected/conservation areas, or departments. Data are mostly sketchy, fragmented, sometimes non-standardised and their scientific robustness is largely questioned. It is therefore a big challenge to obtain data of removed animals to examine the temporal dynamics of control activities during the last decades, particularly at a scale larger than protected areas or departments. However, such analyses are crucial (1) to evaluate whether captures affect coypu and muskrat populations, (2) to examine the spatial dynamics of trapping efforts, and (3) to define areas where trapping efforts should be intensified or 4) whether control strategy should be revised.

The aim of this study is to analyse the temporal and spatial dynamics of AIAR removed by trappers in Pays de la Loire Region during the last 10 years in France. From the data that have been gathered by a network of trappers, we report an estimate of the number of individuals that have been removed per year at the regional level and test whether temporal trends exist in each department during the last decade. We use this data to discuss whether populations of coypu and muskrat are limited or not by the permanent control programme.

## Methods

### Ethics statement

This study analyses data that have been collected by Polleniz. Polleniz is the regional entity recognised by the French Ministry of Agriculture and Fisheries and French Ministry of Ecology and Sustainable Development, decree of the 06/4/2007, as a health organisation in charge of managing the permanent control programme of alien rodents and to collect data from the network of professional actors and local volunteers. Public and private landowners gave permission to trappers to remove AIAR from their property and to conduct this study. The 2924 trappers of the network have obtained an official permit with an individual number or an official agreement to trap from a decree of the local authority including the Prefecture of the five departments Loire-Atlantique, Maine-et-Loire, Mayenne, Sarthe and Vendée (Decrees n˚ 2019/SEE/2194, n˚DAPI-BCC 2007–1179, n˚2019308-001C of the 04/2/2019, n˚07/DDAF/263 of the 16/07/2007 or decree of the 17/2/2016). The trapping procedure compiled with the current laws in France (Decree of the French Ministry of Ecology and Sustainable Development of the 29/1/2007 pursuant to the Article L. 427–8 and 2/9/2016 pursuant to the Article R. 427–6 from the French Environmental Code) and has been fully described in "Trapping procedure" in Methods. Individuals were legally killed and methods used by trappers compiled with the current laws of the French Penal Code pursuant Article 521–1 et R. 654–1. Although an ethical approval from an Ethical Committee is not needed, all procedures for population control of AIAR complied with the ethical standards of the relevant national and European regulations on the care and use of animals (French authority Decision 2007/04/06 and Directive 2010/63/EC). Field studies did not involve endangered or protected species.

### Study area

The study area is Pays de la Loire Region (Fig 1). With a typical oceanic climate, the average annual temperature is 11.5˚C and rainfall is 750mm. Pays de la Loire is an administrative

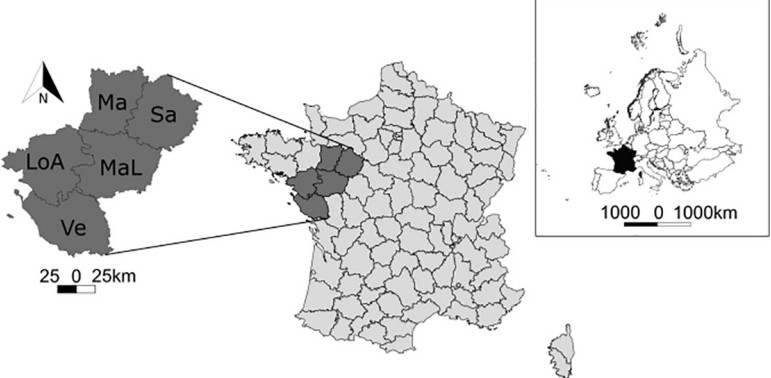

**Fig 1. Location of Pays de la Loire Region in France divided into 5 departments.** LoA is Loire-Atlantique, MaL: Maine-et-Loire, Ma: Mayenne, Sa: Sarthe, and Ve: Vendée. Source of the background map: IGN GEOFLA®.

region of about 32 000 km$^2$ in Western France with 6.3% of its area in wetlands (excluding the Loire River) and hosts 8.7% of the largest wetlands in France. The study area comprises a dense network of rivers and wetlands. It hosts 63 Natura 2000 sites including 16 which are marshes, ponds and lakes of major interest. The Atlantic Ocean borders the region, creating coastal landscapes and marshes with wetlands of major importance (e.g., Lac de Grand-Lieu, Marais Poitevin, La Brière, Le Marais Breton). The relief is rather flat mainly composed of plains and few steep valleys. The average altitude is 78 m with the highest point 417m. The first economic activity is agriculture with 2.2 million hectares of cultivated lands (i.e. 70% of the region). Thus, the landscape is mainly a hedged farmland, meadows, and crops separated by lines of hedges and ditches. The region is divided into five jurisdictional areas called "departments": Loire-Atlantique, Maine-et-Loire, Mayenne, Sarthe, and Vendée (Fig 1) and is composed by 1 238 municipalities with a human density of about 117 residents/km$^2$. The department is the administrative scale at which coypu and muskrat are managed from departmental federations for pest control.

## Control activities

Control activities are coordinated by a regional consortium for invasive species control which involve different types of actors: i.e. trappers, managers, and financers. Trappers could be employed or volunteer, managers are involved in wetland management and conservation, and financers are the local authorities subsidizing control programmes. Polleniz is a regional entity recognised as a health organisation in charge of managing the permanent control programme and to collect data from the network of professional actors and local volunteers.

## Trapping procedure

Coypus and muskrats were captured mainly using one-door cage traps (45 × 45 × 90 cm), X, Conibear trap or two-door cage traps, baited with apples or carrots. The number of cages per volunteer trappers ranged from one to ten but could be higher for professional trappers. All traps have to be checked before noon by law. This is a main factor that limited the number of cages set on the field per trapper. Traps were mainly deployed on the obvious active tracks used by animals near banks of rivers, channels, ponds, ditches, marshes, lakes, collinear restraints, and dams. When banks were abrupt and did not allow traps to be set, they could be set on rafts linked to the banks. Different periods of trapping are commonly used in the programme. Trappers could set traps over (1) several weeks to target a particular area and period, (2) several months to cover their zones moving their traps to different areas particularly when trapper is the sole one in the municipality, (3) one or two seasons that trappers found appropriate to capture during, and (4) the whole year. The summer period (July and August) was not the optimal period of trapping because of vacations and the recreational activities of visiting tourists. Trappers are mostly volunteers although some of them are professional (as is the case in Vendée). The number of volunteer and/or professional trappers varied between municipalities and departments (S1 Table). The number of trappers per municipalities can vary from one to tens depending mostly on the budget allocated to control the species and the level of damage they triggered. The trapping procedure compiled with the current laws in France (Decree of the French Ministry of Ecology and Sustainable Development of the 29/1/2007 pursuant to the Article L. 427–8 and 2/9/2016 pursuant to the Article R. 427–6 from the French Environmental Code).

## Data collection

Trappers, regardless if they are professionals or volunteers, are mandated to complete a notebook of captures, reporting the number of coypus and muskrats removed per day and

municipality. Twice a year, local meetings allowed trappers to communicate their data to their contact in each departmental federation for pest control. We asked the five departmental contacts to communicate their data and we created a regional database containing all captures at the municipality level from 2008 to 2019.

In Sarthe, the total number of AIAR removed did not discriminate between coypus and muskrats, and the number of trappers was not reported (S1 Table). The permanent control programme did not cover all municipalities in Pays de la Loire Region. Over the last 10 years, the percentage of municipalities with available data was on average 68% in Loire-Atlantique, 64% in Maine-et-Loire, 65% in Mayenne, 55% in Vendée, and 32% in Sarthe (S1 Table). The percentages of municipalities with data varied within and between department and year. During the last three years (from 2017 to 2019) the number of data collected by the permanent control programme decreased (S1 Table).

The trapping effort is an important component of any control programme. Unfortunately, local authorities did not initially mandate trappers to record when and how many traps were active in the field. The only available data to evaluate the trapping effort was the number of trappers active per municipality and the number of animals individually removed. Trappers who removed many animals in a municipality might indicate that their trapping effort was high and/or the local population density of AIAR was high. As most trappers did not change their practices (i.e. trapping procedure and effort) between year (personal communication from the contacts of the programme in each departmental federation), we thus believed that the overall number of animals removed per trapper per year was an acceptable index to examine variation in captures between year.

### Data analysis

We fitted a Generalized Least Squares (GLS) model to test for the effect of time (i.e. year) on the total number of coypus and muskrats that had been removed. A common application of GLS estimation is a time-series regression, in which it is generally implausible to assume that errors are independent [41]. We also included the quadratic effect of time (i.e. time$^2$) to investigate a potential curvilinear relationship. To handle autocorrelation in our time series, we ran an autocorrelation function to identify the time lag after which the autocorrelation estimates was confined into the 95% confidence intervals assuming no serial autocorrelation [42]. Then, we implemented in a GLS model an autoregressive moving average (ARMA) term in which the moving average (MA) errors were accounted for using the maximum time lag determined in the previous step [43]. We paid special attention to check the normality of residuals, the distribution of these residuals against fitted values and the lack of sequential autocorrelation in residuals. The same procedure was applied to the total number of coypus, the total number of trappers, the total number of animals of both species, and coypus removed per trapper. Finally, we used the same procedure to analyse the effect of the number of trappers on the total number of animals of both species as well as coypus only. We ran each model in each department separately. Analyses were not performed at a municipality level as spatial autocorrelation existed in our data set (see S1 Fig).

We used QGIS [44] to perform maps and statistical analyses were performed using R 3.6.1 [45], using the nlme package [46].

### Results

First, we investigated the number of AIAR removed per year over the study period (2008–2019). The total number of individuals removed per year increased with time in the Pays de la Loire Region, ranging from 186 438 individuals in 2008 to 287 763 in 2016 (Fig 2A, S1 Table).

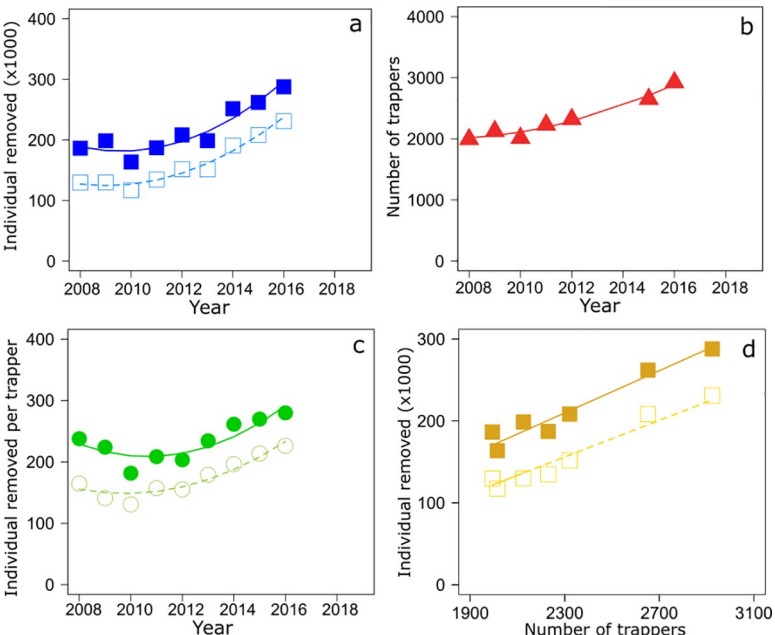

**Fig 2. Temporal dynamics of captures of aquatic invasive alien rodents (AIAR) in the region Pays de la Loire.**
Variation over time in (a) the number of AIAR (full blue square) and coypus only (open square) trapped, (b) the
number of trappers, and (c) the number of AIAR (full green dot) and coypus only (open dot) removed per trappers.
Variation in (d) of the number of AIAR (full golden square) and coypus only (open square) trapped with the number
of trappers. Curved lines represent significant trends from generalized least squares (GLS) models with an
autoregressive moving average term (see S2 and S3 Tables for statistical details).

Although about 80% of AIAR removed were coypus (S1 Table), analyses showed that the number of AIAR and coypus only removed increased by 50% in the region (Fig 2A) and in each department (about 108% increase in Loire-Atlantique, 54% in Mayenne, 52% in Sarthe, 39% in Maine-et-Loire, and 35% in Vendée, Figs 3 and 4) with a nonlinear way (S2 Table), although data were available only for two departments beyond 2016.

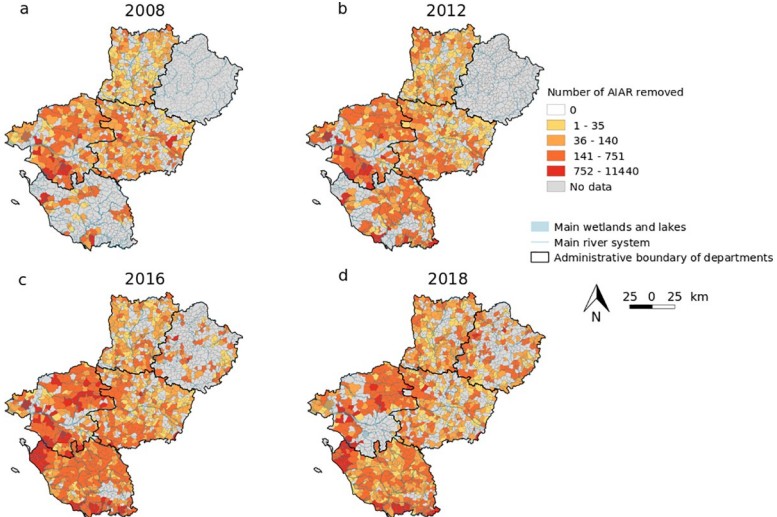

**Fig 3. Number of aquatic invasive alien rodents (AIAR) removed per municipality in the region Pays de la Loire.**
In (a) 2008, (b) 2012, (c) 2016 and (d) 2018 with no data available at a municipality level in Sarthe in the three former
periods (see S1 Table for details). Source of the background map: IGN GEOFLA®.

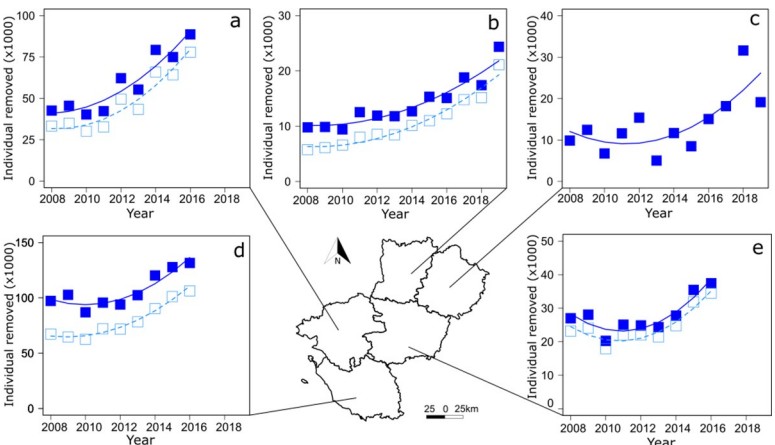

**Fig 4. Variation over time of the number of aquatic invasive alien rodents (AIAR) in the 5 departments of the region Pays de la Loire.** AIAR (full square) and coypus only (open square) removed in (a) Loire-Atlantique, (b) Mayenne, (c) Sarthe, (d) Vendée, (e) Maine-et-Loire. Curved lines represent significant trends from generalized least squares (GLS) models with an autoregressive moving average term (see S2 Table for statistical details). Source of the background map: IGN GEOFLA®.

During the same time period, the number of trappers increased from 1 994 in 2008 to 2 924 in 2016 (i.e. 50% increase) in the Pays de la Loire Region (Sarthe not considered because of missing data, see S1 Table) (Fig 2B). The number of trappers increased linearly in Mayenne (Fig 5B) and in a curvilinear trend in Loire-Atlantique (Fig 5A) and Maine-et-Loire (Fig 5D) although it remained constant in Vendée (Fig 5C, S2 Table). Apart from Vendée (Fig 6C), the number of captures increased with the rise in the number of trappers in the Pays de la Loire Region (Fig 2B), in Loire-Atlantique (Fig 6A), Mayenne (Fig 6B), and Maine-et-Loire (Fig 6D, S3 Table).

Since the increase of AIAR trapped over time could be explained by an increase in the number of trappers, we examined how the number of animals removed per trapper changed with time. Discarding Sarthe due to missing data (S1 Table), the ratio between the total number of AIAR removed and the number of trappers indicated that each trapper removed about 92

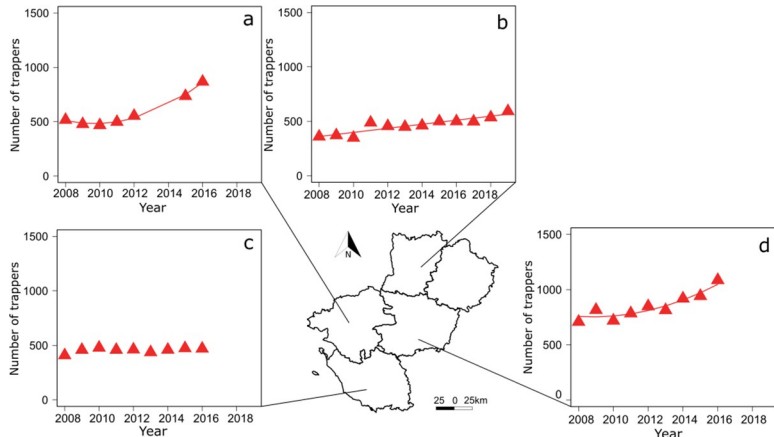

**Fig 5. Variation over time in the number of trappers in 4 departments of the region Pays de la Loire.** In (a) Loire-Atlantique, (b) Mayenne, (c) Vendée, and (d) Maine-et-Loire. Data in Sarthe were not available. Curved lines represent significant trends from generalized least squares (GLS) models with an autoregressive moving average term (see S2 Table for statistical details). Source of the background map: IGN GEOFLA®.

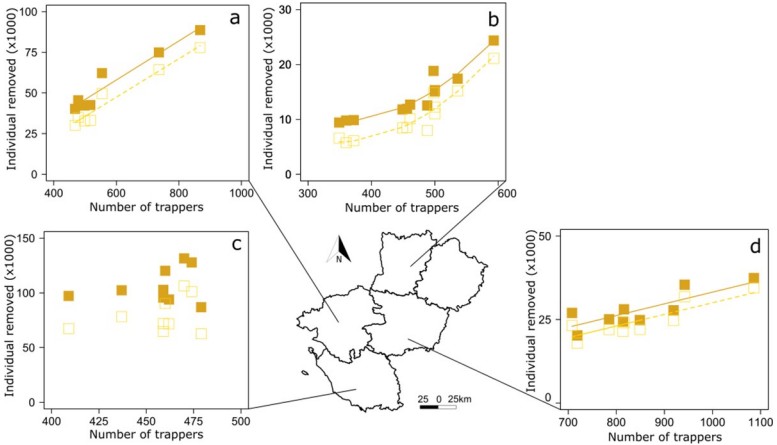

**Fig 6. Variation of the number of aquatic invasive alien rodents (AIAR) with the number of trappers in the 5 departments of the region Pays de la Loire.** AIAR (full square) and coypus only (open square) removed in (a) Loire-Atlantique, (b) Mayenne, (c) Vendée, and (d) Maine-et-Loire. Data in Sarthe were not available. Curved lines represent significant trends from generalized least squares (GLS) models with an autoregressive moving average term (see S3 Table for statistical details). Source of the background map: IGN GEOFLA®.

individuals per year in the region Pays de la Loire. In regards to the departments, the same assessment in 2016 (except Sarthe) highlighted that the number of AIAR (and coypus in brackets) removed per trapper reached about 280 (226) in Vendée, 102 (90) in Loire-Atlantique, 35 (32) in Maine-et-Loire, and 30 (24) in Mayenne (S1 Table). Although we did not detect a significant change over time in the total number of AIAR removed per trapper at the regional level, the number of coypus removed per trapper increased by 22% over time (Fig 2C, S2 Table). Analyses showed that the number of AIAR or coypus only removed per trapper increased with time in at least 3 departments (Mayenne (Fig 7B), Vendée (Fig 7C) and Loire-Atlantique (Fig 7A) with the two former showing a curvilinear trend, see S2 Table). Although we detected a curvilinear variation in the number of trappers over time in Maine-et-Loire (Fig 7D, S2 Table), analyses showed that the number individuals removed per trapper remained similar in 2008 and 2016.

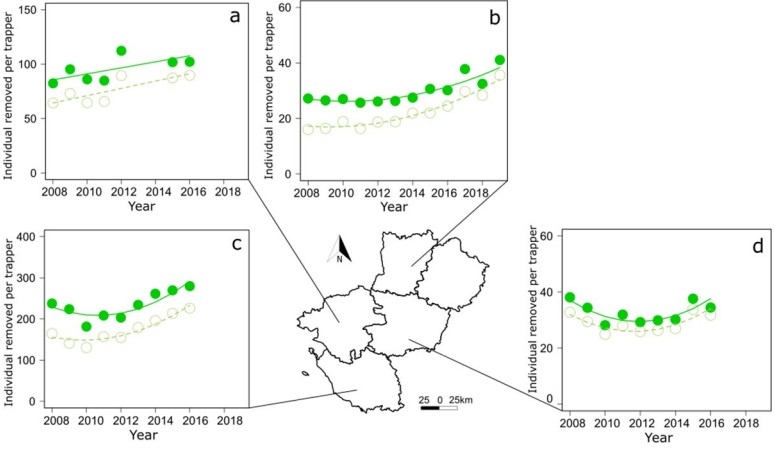

**Fig 7.**

## Discussion

Although the database from the permanent control programme was sometimes heterogeneous across municipalities and departments, our study is unique. The total number of AIAR removed per year in Pays de la Loire Region (32 000 km$^2$ with 6.3% of wetlands) reached an outstanding amount of about 288 000 individuals (about 20% muskrats and 80% coypus) in 2016. To make a comparison, the programme covering 23 384 km$^2$ in South Korea managed to trap 27 487 coypus in 5 years-period (2014–2018) [18], while in Italy in an area of 41 515 km$^2$ more than 64 000 coypus were removed in a year [16]. In East Anglia (UK) 34 822 coypu were removed from 5 379 km$^2$ to eradicate the species [47]. The number of AIAR removed by the programme here would represent about 70% (424 907) of the total AIAR shot during the game season at the national level in France [40].

In 10 years, the total number of AIAR removed per year increased by 50% in the region. During the same time length, the number of trappers involved in the programme each year increased also by 50%. Thus, the strength in the increase of AIAR trapped over time might be explained by the increase in the number of active trappers and it might explain why we did not detect any significant change with time in the total number of AIAR removed per trapper at the regional level. However, the number of coypus removed per trapper increased by 22%, and the number of AIAR or coypus only removed per trapper increased at least in 3 departments including Mayenne, Vendée, and Loire-Atlantique. Thus, these increases of both the (1) AIAR and coypus only removed in these departments and (2) the number of coypus removed in the region did not seem to support the hypothesis that the increase of captures was mainly explained by the increase of the trapping effort with time. Although we acknowledged that variation of captures with time should be evaluated, controlling for the number of trap-days [48]– information that the programme did not collect–the increase in the total number of AIAR removed per trapper per year over time suggest that population of coypus is still not limited by control activities; on the contrary, it is probably increasing.

Over the last decade, the spatial dynamics of captures was heterogeneous although the number of municipalities involved in the programme varied across department. The number of AIAR removed per department increased from 35% in Vendée to 108% in Loire-Atlantique. This difference between departments might be explained by several factors including variation in the local density of AIAR, the amount of suitable habitats, the number of trappers (volunteers and professionals), and the local management plan of municipalities to encourage AIAR trapping (including rewards). Future studies should thus investigate factors influencing temporal dynamics of captures at a municipality level.

The permanent control programme in Pays de la Loire Region is mainly based on trappers who are mainly volunteers and thus makes this programme unique. One of the strategies that has been used to promote recruitment of volunteer trappers was to reward them for their AIAR captures. Rewards are locally defined (municipalities/departments) and are not the same across all municipalities in the region. Studies have questioned whether rewards might not alter the performance of trappers when they are perceived as an extra salary [49]. The reward in Pays de la Loire is on average 1.5 euro/animal and though there are several trappers which caught hundreds of individuals per year, the overall reward is still limited. Although the reward is an important (but not unique) argument to recruit new volunteers and to maintain the ones who are involved in the programme, it might also adversely lead trappers to maintain a population of coypus when it is a significant bonus per year. We believe that future studies should (1) test the efficiency and sustainability of such strategy in a permanent control programme and (2) examine motivations of trappers to be part of the programme.

The number of data gathered from the programme decreased since 2017. Several explanations can be proposed here: (1) Several municipalities have decreased their funds and withdrew from AIAR control. For instance, some municipalities have stopped rewarding trappers and the number of trappers declined in some areas (apart from Vendée). (2) Subsidies from departments towards AIAR control declined, affecting the consortium to manage the permanent control programme. This is for instance the case when 20% of municipalities gave up the programme in Loire-Atlantique in 2017 and 2018 (S1 Table, Fig 3D) or when the programme lost its departmental contact in Sarthe. (3) The cost of the permanent control programme is very expensive [17, 18] and the programme has to deal with a local decline in annual willingness to pay for AIAR control. (4) during the last few years, all data were not inserted in the regional database by departmental federations.

A strategy that should encourage decision makers and financers to fund the permanent control programme is to assess the direct or indirect costs related to the presence of AIAR in ecosystems [50, 51]. One of the decisive arguments to convince local, federal, or national governmental authority is that AIAR are healthy carriers of significant diseases which are contagious to humans including leptospirosis, toxoplasmosis, and alveolar echinococcosis [24]. In France, about 600 cases of leptospirosis were confirmed in humans in 2014 for which 75% of cases were due to the use of wetlands through recreational-based water activities [52]. For instance, a cluster of seven kayakers on the river Vilaine in Britany (France) with clinical symptoms of leptospirosis was reported to French health authorities in 2016 [53]. Although we did not find any study assessing costs of medical care, we predict that they would be significantly high if the number of contagious events increases. Whilst digging their burrows, coypus and muskrats make riverbanks unstable and sometimes may collapse. Costs associated with hydraulic damages related to the need to clean streams have been documented to be important [16, 28]. Indirect damages on hydraulic structures including dams may cause flooding, leading to other local costly damages. Agricultural damages on crops near streams and marshes in which AIAR have established have been documented [28, 54]. Although studies are rare, refunding farmers due to agricultural damages of AIAR might be very costly [55]. Furthermore, it has been documented that AIAR affect native species in different taxa, for instance plants [35, 37], birds [30], or molluscs [31], their direct and indirect effects on population dynamics of native species and more generally on ecosystem functions have rarely been studied. Despite the fact that it is commonly accepted that alien species contribute to the transformation and downgrading of ecosystems, more studies are needed to assess the costs of habitat restoration and more generally on the loss of ecosystem services provided by wetlands where AIAR have established communities. Finally, although it is not common in our studied area, other initiatives of fundraising/sponsoring potential partnerships with private companies should be explored to strengthen the permanent control programme against AIAR.

To conclude, the permanent control programme has managed 1) to remove an outstanding number of AIAR per year in the region Pays de la Loire and 2) to recruit new volunteer trappers during the last decades. Despite these successes, our analyses suggest that control activities did not limit the population of AIAR and the number of data gathered decreased since 2017. Our study pleads for a discussion between all stakeholders including actors involved in this programme, local, and regional authorities and decision makers, scientists, and citizens to decide collectively objectives of captures and priority zones and to encourage financers to fund this permanent control programme. Finally, the permanent control programme would benefit from a scientific study monitoring population dynamics and crucial biological parameters of AIAR including, survival, reproduction, and migratory rates.

## Supporting information

**S1 Fig. Correlogram of the variation of the Moran's I index assessed from the number of captures per municipalities.** Open dot indicates that Moran's I index is not significant. Red full dot indicates a significant Moran's index with a value being outside the 95% CI assessed from 1000 simulations. This analysis has been run from all data of AIAR captures in 2016 using *pgirmess* package in R. This figure shows that the number of captures in a municipality is indisputably spatially correlated with captures in neighbouring municipalities distant up to 40 km and in a lesser extent up to 90 km.
(DOCX)

**S1 Table. Data that were available per year and department in the permanent control programme on the Alien Invasive Aquatic Rodents (AIAR) in the region Pays de la Loire.**
(DOCX)

**S2 Table. Statistics from the generalized least squares models with Auto Regressive Moving Average term (ARMA (q = 1)) of the effect of time on the number of Alien Invasive Aquatic Rodents (AIAR) removed (coypus and muskrats), coypus only, number of trappers, AIAR removed per trapper and number of municipalities that encountered positive value of inter-annual captures rates of animals (both coypu and muskrat and coypu only).** Each GLS run is presented with the F-value, the numDF, the p-value and the Pseudo-$R^2$.
(DOCX)

**S3 Table. Statistics from the generalized least squares models with Auto Regressive Moving Average term (ARMA (q = 1)) of the effect of the number of trappers on the number of Alien Invasive Aquatic Rodents (AIAR) removed (coypus and muskrats) and coypus only, number of trappers, AIAR removed per trapper, and number of municipalities that encountered positive value of inter-annual captures rates of animals (both coypu and muskrat and coypu only).** Each GLS run is presented with the F-value, the numDF, the p-value and the Pseudo-$R^2$.
(DOCX)

## Acknowledgments

We would like to thank all regional and local partners including volunteers who contribute to make this study possible. Finally, we thank Franck Courchamp for his helpful references on the economic costs on alien species and Marie-Alice Budniok, Jürgen Tack, and Martin Fox from European Landowners' Organization for their helpful comments and English editing.

## Author Contributions

**Conceptualization:** Manon Bonnet, Olivier Pays.

**Data curation:** Manon Bonnet.

**Formal analysis:** Manon Bonnet, Olivier Pays.

**Methodology:** Olivier Pays.

**Supervision:** Olivier Pays.

**Writing – original draft:** Manon Bonnet, Olivier Pays.

**Writing – review & editing:** Manon Bonnet, Gérald Guédon, Marc Pondaven, Sandro Bertolino, Damien Padiolleau, Vanessa Pénisson, Francine Gastinel, Fabien Angot, Pierre-Cyril Renaud, Antonin Frémy, Olivier Pays.

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
