## [Decision Letter · Decision Letter 0]

25 Jan 2021

PONE-D-20-36473

Aquatic invasive alien rodents in western France: where do we stand today after decades of control?

PLOS ONE

Dear Dr. Pays,

Thank you for submitting your manuscript to PLOS ONE. After careful consideration, we feel that it has merit but does not fully meet PLOS ONE’s publication criteria as it currently stands. Therefore, we invite you to submit a revised version of the manuscript that addresses the points raised during the review process.

We look forward to receiving your revised manuscript.

Kind regards,

Bi-Song Yue, Ph.D

Academic Editor

PLOS ONE

'This is an unfunded study. The authors received no specific funding for this study.' 

We note that one or more of the authors are employed by a commercial company: Polleniz.

3. We note that Figures 1, 3, 4, 5 and 6 in your submission contain map images which may be copyrighted.

a. You may seek permission from the original copyright holder of Figures 1, 3, 4, 5 and 6 to publish the content specifically under the CC BY 4.0 license. 

Reviewers' comments:

Reviewer's Responses to Questions

**Comments to the Author**

1. Is the manuscript technically sound, and do the data support the conclusions?

Reviewer #1: Yes

Reviewer #2: Yes

2. Has the statistical analysis been performed appropriately and rigorously? 

Reviewer #1: Yes

Reviewer #2: Yes

3. Have the authors made all data underlying the findings in their manuscript fully available?

Reviewer #1: Yes

Reviewer #2: Yes

4. Is the manuscript presented in an intelligible fashion and written in standard English?

Reviewer #1: Yes

Reviewer #2: Yes

5. Review Comments to the Author

Reviewer #1: General comment

The manuscript presents a very interesting work about the difficult field management of two invasive mammal species in a large area in northwest France. It was quite impressive the amount of data (individuals captured and traps) and the time period analyzed (10 years), in addition to the people working in the field to make the captures. The results reach the proposed objectives, which was to analyze the temporal and spatial dynamics of captures by trappers involved in this program during the last 10 years to evaluate its effectiveness. The analyses seemed appropriate to the proposal of the work and in fact support the conclusions. However, some points caught my attention and I explain them in sections, as follows.

Abstract

Concise and very informative, the summary makes the work and its results very clear. However, I think that the end of the abstract can be used to list some final points of the conclusions, because the way it is written does not make it clear what are the 'several explanations'.

Introduction

The introduction is relatively short, but very objective, citing key articles in the literature on species invasion and control of invasive species. The chain of ideas presented allows readers to understand the context of the problem situation, especially in relation to specific citations of invasions of these species in France. However, I do not know if there is a lack of literature, but I have not seen results on the impacts of these invasions in the study region, so, I think it is very interesting to have one more paragraph before the objectives, to add information about other invasions or even work related to the invasions of these two species in the specific region of Pays de la Loire Region.

Methods

The methods are very well written and detailed. However, I suggest changing the subtitle only to 'Methods'. The subsection 'Study area' is very well written, and in Figure 1, I suggest adding a scale bar to the central map, I know that this is not a rule either, but adding a compass with the north can be interesting. Nothing to add in the subsections 'Control activities', 'Trapping procedure', and 'Data collection'. In the subsection 'Data analysis' I have some points to list:

1. Why did the authors not use a GLMM (with Poisson family), with the fixed part of the model being the temporal relationship to analyze the data, in addiction with model selection?

2. It would also be interesting to analyze the number of Trappers as a predictor for Number of AIAR removed and Number of coypus removed, analyzing the years as a fixed factor, to answer whether in fact the reduction of the traps influences the Number of AIAR and coypus

3. The authors present a very interesting result in figure 3, which are the catches by municipalities over time. Suggest doing a spatial analysis (Moran's I index global and local) to analyze the spatial structuring of the catches and to know if the populations are grouped over time

4. If there are annual municipal data, it would be very interesting to do these regressions by municipalities. Sometimes patterns can arise from the grouping of data, which are often not observed for the ungrouped data

5. Lastly, the authors also mention the effects of AIAR on several points: public health (leptospirosis), damages on hydraulic structures and affect native species in different taxa. A very interesting analysis would be to compile this information at the municipal level: cases of leptospirosis, drainage density and occurrences of native species affected by AIAR and to make the spatial relationship to assess the relationship of the number of AIAR with these factors. Thus, since resources are scarce to control these species, this analysis could indicate priority locations for mobilizing more effective control

6. These analyze could be combined with a study of payment for environmental services, to translate in a monetized way, how much the control program reduced the impacts and how much it benefited in the reduction of losses caused by invasive species.

Results

The results are concise and very written and explained. However, I suggest improving the appearance of the regressions. One possibility is to use different colors to discriminate the number of AIAR and coypus only, in addition to increasing the formats that represent the data and the regression lines.

Discussion

The discussion focuses on analyzing the results in order to compare them with similar results from other works.

On lines 275-276 "Studies have questioned whether rewards might not alter the performance of trappers when they are perceived as an extra salary", I missed the references to support this statement.

In the penultimate paragraph, the authors cite the decline in data collection since 2017, attributing mainly to the lack of resources for the control program. However, I missed possible partnerships with private sources to increase fundraising for this purpose. I don't know how these initiatives work in France, and especially in the region, but it can be a way out of the lack of resources.

Finally, the authors show that a strategy to obtain resources from decision makers is the impacts on public health (leptospirosis), damages on hydraulic structures and affect native species, ending with the question of more studies to evaluate the costs and ecosystem services, but here I missed a conclusion for the proposed objectives: "where do we stand today after decades of control" and "We use this data to discuss whether populations of coypu and muskrat are limited or not by the permanent control program.", or that is, did the analyses in fact make it possible to analyze the state of these regions in relation to the control program? And the populations were actually limited by the control program?

Reviewer #2: This study is to analyse the temporal and spatial dynamics of AIAR removed by trappers in Pays de la Loire Region during the last 10 years in France. From the data that have been gathered by a network of trappers, researchers report an estimate of the number of individuals that have been removed per year at the regional level and test whether temporal trends exist in each department during the last decade. They use this data to discuss whether populations of coypu and muskrat are limited or not by the permanent control programme.

However, there still are some concerns need to be clarified

1.In some areas, the total number of AIAR removed did not discriminate between coypus and muskrats, and the number of trappers was not reported. I suggested to improve relevant data.

2.The experimental design is too simple. Is the number of AIAR removed by trappers affected by other factors? For example, some natural factors.

3.The total number of AIAR in the area each year and the number of AIAR migrated into this area are not considered. These data are also very important for the evaluation of removal efficiency.

6. PLOS authors have the option to publish the peer review history of their article (what does this mean?). If published, this will include your full peer review and any attached files.

Reviewer #1: No

Reviewer #2: No

---

## [Author Response · Author response to Decision Letter 0]

24 Mar 2021

Reviewer's Responses to Questions

Comments to the Author

1. Is the manuscript technically sound, and do the data support the conclusions?

Reviewer #1: Yes

Reviewer #2: Yes

2. Has the statistical analysis been performed appropriately and rigorously?

Reviewer #1: Yes

Reviewer #2: Yes

3. Have the authors made all data underlying the findings in their manuscript fully available?

Reviewer #1: Yes

Reviewer #2: Yes

4. Is the manuscript presented in an intelligible fashion and written in standard English?

Reviewer #1: Yes

Reviewer #2: Yes

5. Review Comments to the Author

Referee #1

The manuscript presents a very interesting work about the difficult field management of two invasive mammal species in a large area in northwest France. It was quite impressive the amount of data (individuals captured and traps) and the time period analyzed (10 years), in addition to the people working in the field to make the captures. The results reach the proposed objectives, which was to analyze the temporal and spatial dynamics of captures by trappers involved in this program during the last 10 years to evaluate its effectiveness. The analyses seemed appropriate to the proposal of the work and in fact support the conclusions. However, some points caught my attention and I explain them in sections, as follows.

RESPONSE: Many thanks for the positive comments. Please find below the responses to your comments.

Abstract

Concise and very informative, the summary makes the work and its results very clear. However, I think that the end of the abstract can be used to list some final points of the conclusions, because the way it is written does not make it clear what are the 'several explanations'.

RESPONSE: As suggested, we have tried to revise the last two sentences of the abstract considering the words limit. We have now changed “We addressed here several explanations including the decline of subsidies allocated towards collective fights against alien aquatic rodents” mentioning “Indeed, although rewards are crucial to recruit new volunteers, subsidies from local and regional authorities are declining. Decision makers and financers should be encouraged to fund this programme from the perspectives of the direct or indirect costs related to the presence of aquatic invasive alien rodents in wetlands.” This is now lines L40-44.

Introduction

The introduction is relatively short, but very objective, citing key articles in the literature on species invasion and control of invasive species. The chain of ideas presented allows readers to understand the context of the problem situation, especially in relation to specific citations of invasions of these species in France. However, I do not know if there is a lack of literature, but I have not seen results on the impacts of these invasions in the study region, so, I think it is very interesting to have one more paragraph before the objectives, to add information about other invasions or even work related to the invasions of these two species in the specific region of Pays de la Loire Region.

RESPONSE: This is a good point but unfortunately there is a lack of literature. As we mentioned in the introduction, there are few published studies reporting the current management strategies for these species and their results at the national or regional level. We cited the study from Aubry et al. (2016) reporting in 2016 that around 350 000 coypus and 70 000 muskrats were shot during the 2013-2014 game season in France. We did not find any other articles documenting the number of captures or the impacts of these species in the region Pays de la Loire or at the national level. Few studies that have been published in France in the last decades on the effect of AIAR. This is clearly stated in our article.

As we mentioned in the text, unpublished data of captures exist but they are mostly sketchy, fragmented, sometimes non-standardised, and their scientific robustness is largely questioned. Thus, in this article, we wanted to refer to studies that have been peer reviewed by the scientific experts. 

However, to consider referee’s suggestion, we have now included these new references to the article:

Bos D, Kentie R, La Haye M, Ydenberg RC. Evidence for the effectiveness of controlling muskrat (Ondatra zibethicus L.) populations by trapping. Eur J Wildl Res. 2019;65: 45. (L69)

Gethöffer F, Siebert U. Current knowledge of the Neozoa Nutria and Muskrat in Europe and their environmental impacts. J Wildl Biodivers. 2020;4: 1-12. (L69)

Guillois Y, Bourhy P, Ayral F, Pivette M, Decors A, Aranda Grau JH, et al. An outbreak of leptospirosis among kayakers in Brittany, North-West France, 2016. Euro Surveill. 2018;23: 1700848. (L338)

Jouventin P, Micol T, Verheyden C, Guédon G. Le Ragondin. Biologie et méthodes de limitation des populations. Paris: Editions ACTA, France. 1996. (L85)

Methods

The methods are very well written and detailed. However, I suggest changing the subtitle only to 'Methods'. 

RESPONSE: Corrected as suggested (L104)

The subsection 'Study area' is very well written, and in Figure 1, I suggest adding a scale bar to the central map, I know that this is not a rule either, but adding a compass with the north can be interesting. Nothing to add in the subsections 'Control activities', 'Trapping procedure', and 'Data collection'. 

RESPONSE: Fig 1 has been revised as suggested (L142)

In the subsection 'Data analysis' I have some points to list:

1. Why did the authors not use a GLMM (with Poisson family), with the fixed part of the model being the temporal relationship to analyze the data, in addiction with model selection?

RESPONSE: Variation of the number of captures or trappers over time was investigated using statistical procedure to analyse time series, which was our main concern. A common application of generalized least-squares (GLS) estimation is a time-series regression, in which it is generally implausible to assume that errors are independent (Cryer JD, Chan KS. Time series analysis: With applications in R. New York, NY: Springer. 2008.r). This method is for instance commonly used to analyse precipitation data (Pearson & Carroll 1998 Conservation Biology 12: 809–821). As we stated in the methods section, to control for the time-dependence of regression errors, we fitted a first-order auto-regressive model using ‘correlation=corAR1’ function (Zuur AF, Leno EN, Walker NJ, Saveliev AA, Smith GM. Mixed effects models and extensions in ecology with R. New York, NY: Springer. 2009.). As we checked the normality of residuals, the distribution of these residuals against fitted values and the lack of sequential autocorrelation in residuals, we believe that our procedure is fully robust here.

To consider the referee’s concern, we now state (L200-201) that “A common application of GLS estimation is a time-series regression, in which it is generally implausible to assume that errors are independent (Cryer & Chan 2008)” refer to Zuur et al. 2009 and add these two new references to our references list. We hope that these details in the existing paragraph give enough evidence that our procedure is robust.

2. It would also be interesting to analyze the number of Trappers as a predictor for Number of AIAR removed and Number of coypus removed, analyzing the years as a fixed factor, to answer whether in fact the reduction of the traps influences the Number of AIAR and coypus

RESPONSE: This is a really good point. Using the same procedure as described above (i.e. considering time-dependence of regression errors), we ran models testing for the effect of the number of trappers (including also the quadratic term) on the number of AIAR and coypus. Results are presented in a supplementary material S3 Table. Our analyses show that the number of captures increased with the number of trappers at the regional level (Fig 2d). This is also true for 3 departments except in Vendée (Fig 6). Thus, our analyses show that when the trapping effort increases, the number of captures increases. 

We have revised the text to include these new analyses (L248-250). We now mention in the results section “Apart from Vendée (Fig 6c), the number of captures increased with the rise in the number of trappers in the Pays de la Loire Region (Fig 2b), in Loire-Atlantique (Fig 6a), Mayenne (Fig 6b), and Maine-et-Loire (Fig 6d, S3 Table)”. 

3. The authors present a very interesting result in figure 3, which are the catches by municipalities over time. Suggest doing a spatial analysis (Moran's I index global and local) to analyze the spatial structuring of the catches and to know if the populations are grouped over time.

RESPONSE: The aim of this article was to analyse temporal trends of the number of captures and trappers at a regional level and for each department as it is the spatial level at which the permanent control programme against AIAR is organised in our studied area. Moreover, analysing captures at a municipality level requires performing statistical analyses that cope with spatial autocorrelation. 

As suggested by the referee, we analyse the spatial structures of captures using Moran’s I index in 2016 as it is the year for which the permanent control programme has gathered the maximum amount of data. Analyses show that the number of captures in a municipality is indisputably spatially correlated with captures in neighbouring municipalities distant up to 40 km and in a lesser extend up to 90 km. This is not so surprising as AIAR are established all over the region Pays de la Loire.

S1 Fig. Correlogram of the variation of the Moran’s I index assessed from the number of captures per municipalities. 

Open dot indicates that Moran’s I index is not significant. Red full dot indicates a significant Moran’s index with a value being outside the 95% CI assessed from 1000 simulations. This analysis has been run from all data of AIAR captures in 2016 using pgirmess package in R. This figure shows that the number of captures in a municipality is indisputably spatially correlated with captures in neighbouring municipalities distant up to 40 km and in a lesser extend up to 90 km. 

We have added this analysis in supplementary material S1 Fig. This analysis was cited in methods (L213-214) “Analyses were not performed at a municipality level as spatial autocorrelation existed in our data set (see S1 Fig)”. 

4. If there are annual municipal data, it would be very interesting to do these regressions by municipalities. Sometimes patterns can arise from the grouping of data, which are often not observed for the ungrouped data.

RESPONSE. Data of captures have been collected from more than 1500 municipalities in our studied area. As we said in the previous comment, the aim of this article was to analyse the dynamics of captures at the scale from which the permanent control programme operates. Analysing temporal trend for the 1500 municipalities is tedious. Moreover, analysing temporal trends for a given municipality requires to consider the captures of neighbouring municipalities (see previous comment on the spatial autocorrelation existing in the data set). Thus, we think that this interesting idea should be considered in another article. This is now mentioned in discussion “Future studies should thus investigate factors influencing temporal dynamics of captures at a municipality level” (L306-307). 

5. Lastly, the authors also mention the effects of AIAR on several points: public health (leptospirosis), damages on hydraulic structures and affect native species in different taxa. A very interesting analysis would be to compile this information at the municipal level: cases of leptospirosis, drainage density and occurrences of native species affected by AIAR and to make the spatial relationship to assess the relationship of the number of AIAR with these factors. Thus, since resources are scarce to control these species, this analysis could indicate priority locations for mobilizing more effective control.

RESPONSE. This is a very interesting idea. Unfortunately, to our knowledge, such data set does not exist, and we cannot perform this analysis.

6. These analyze could be combined with a study of payment for environmental services, to translate in a monetized way, how much the control program reduced the impacts and how much it benefited in the reduction of losses caused by invasive species.

RESPONSE: We agree with the referee, it would be a fascinating project. However, from the data that are available, it is not possible to run this analysis. As we mentioned in the discussion section, the direct or indirect costs related to the presence of AIAR in our ecosystems are hard to assess. Here again, we think that it should be the focus of another article. 

Results

The results are concise and very written and explained. However, I suggest improving the appearance of the regressions. One possibility is to use different colors to discriminate the number of AIAR and coypus only, in addition to increasing the formats that represent the data and the regression lines.

RESPONSE: Corrected as suggested (L227-233, L238-243, L251-255, and L256-261)

Discussion

The discussion focuses on analyzing the results in order to compare them with similar results from other works.

On lines 275-276 "Studies have questioned whether rewards might not alter the performance of trappers when they are perceived as an extra salary", I missed the references to support this statement.

RESPONSE: We have added this new reference (L313).

Sheail, J. The grey squirrel (Sciurus carolinensis)—a UK historical perspective on a vertebrate pest species. J Environ Manage, 1999;55: 145-156. 

In the penultimate paragraph, the authors cite the decline in data collection since 2017, attributing mainly to the lack of resources for the control program. However, I missed possible partnerships with private sources to increase fundraising for this purpose. I don't know how these initiatives work in France, and especially in the region, but it can be a way out of the lack of resources.

RESPONSE: In France, most of the funds allocated to the permanent control programme of AIAR comes from the departments or other local authorities such as groups of municipalities or local watershed unions. Local hunting associations, municipalities, or other local public administrations that have an interest in rivers or artificial water damming on their territories can also contribute financially to this programme. However, as we mention in the text, the programme exists because it manages to recruit volunteer trappers. To respond to the referee’s suggestion, private companies do not contribute to this programme for different reasons: 1) It is an initiative that has not been fully explored by the programme, 2) private companies might have ethical concerns to be involved in such programme, 3) it is commonly assumed that captures should be managed by local authorities.

To consider the referee’s thinking, we mention this potential initiative in the discussion (L351-353): “Finally, although it is not common in our studied area, other initiatives of fundraising sponsoring potential partnerships with private companies should be explored to strengthen the permanent control programme against AIAR.” 

Finally, the authors show that a strategy to obtain resources from decision makers is the impacts on public health (leptospirosis), damages on hydraulic structures and affect native species, ending with the question of more studies to evaluate the costs and ecosystem services, but here I missed a conclusion for the proposed objectives: "where do we stand today after decades of control" and "We use this data to discuss whether populations of coypu and muskrat are limited or not by the permanent control program.", or that is, did the analyses in fact make it possible to analyze the state of these regions in relation to the control program? And the populations were actually limited by the control program?

RESPONSE: A suggested, we added a last paragraph of conclusion in the discussion. We now mention (L354-362): “To conclude, the permanent control programme has managed 1) to remove an outstanding number of AIAR per year in the region Pays de la Loire and 2) to recruit new volunteer trappers during the last decades. Despite these successes, our analyses suggest that control activities did not limit the population of AIAR and the number of data gathered decreased since 2017. Our study pleads for a discussion between all stakeholders including actors involved in this programme, local, and regional authorities and decision makers, scientists, and citizens to decide collectively objectives of captures and priority zones and to encourage financers to fund this permanent control programme. Finally, the permanent control programme would benefit from a scientific study monitoring population dynamics and crucial biological parameters of AIAR including, survival, reproduction, and migratory rates.” 

Referee #2: 

This study is to analyse the temporal and spatial dynamics of AIAR removed by trappers in Pays de la Loire Region during the last 10 years in France. From the data that have been gathered by a network of trappers, researchers report an estimate of the number of individuals that have been removed per year at the regional level and test whether temporal trends exist in each department during the last decade. They use this data to discuss whether populations of coypu and muskrat are limited or not by the permanent control programme.

However, there still are some concerns need to be clarified

1.In some areas, the total number of AIAR removed did not discriminate between coypus and muskrats, and the number of trappers was not reported. I suggested to improve relevant data.

RESPONSE: Yes, it is true, and this lack of data exists for trappers operating in the Sarthe (one of the 5th department). However, we found a successful way to analyse the temporal trends of captures of AIAR in this department and at a regional scale. Thus, to respond to the referee’s concern, we cannot improve the data set when data do not exist.

2.The experimental design is too simple. Is the number of AIAR removed by trappers affected by other factors? For example, some natural factors.

RESPONSE: As we mentioned in the text, this study investigating temporal trends of captures and trappers at a regional scale is unique in France. Indeed, there is no other comparable study in France. The aim of this study is the analysis temporal trends during the last decades to provide an assessment of the permanent control programme operating mostly with volunteer trappers. Thus, the effect of the natural factors on the captures is beyond the scope of our article.

With that said, we agree that lots of factors might influence the number of AIAIR removed by trappers including environmental factors. For instance, figure 3 shows that reservoirs of AIAR in our studied areas are mainly located in the Western part of the region. Indeed, these two departments, Loire-Atlantique and Vendée, host the largest wetlands, marshes, and lakes of Pays de la Loire region. Although it was mentioned in Methods, we have revised our text to make this clearer. We now mention (L132-133): “The Atlantic Ocean borders the region, creating coastal landscapes and marshes with wetlands of major importance (e.g., Lac de Grand-Lieu, Marais Poitevin, La Brière, Le Marais Breton)”. 

3.The total number of AIAR in the area each year and the number of AIAR migrated into this area are not considered. These data are also very important for the evaluation of removal efficiency.

RESPONSE. Yes, we agree that such data are important. More broadly, a scientific project investigating population dynamics would be useful in the permanent control programme to assess crucial biological parameters of survival and reproduction. However, there is no such ongoing project in the Pays de la Loire region and in its surrounding. In France, data on biological performance of AIAR are old (they are over 30 years old, see Jouventin et al. 1996 cited in our article (L85)) and should be updated. To consider the referee’s thinking, we have now mentioned in the Discussion (L351-353): “Finally, the permanent control programme would benefit from a scientific study monitoring population dynamics and crucial biological parameters of AIAR including, survival, reproduction, and migratory rates.” 

6. PLOS authors have the option to publish the peer review history of their article (what does this mean?). If published, this will include your full peer review and any attached files.

Do you want your identity to be public for this peer review? For information about this choice, including consent withdrawal, please see our Privacy Policy.

Reviewer #1: No

Reviewer #2: No

---

## [Editor Report · Decision Letter 1]

29 Mar 2021

Aquatic invasive alien rodents in western France: where do we stand today after decades of control?

PONE-D-20-36473R1

Dear Dr. Pays,

We’re pleased to inform you that your manuscript has been judged scientifically suitable for publication and will be formally accepted for publication once it meets all outstanding technical requirements.

Kind regards,

Bi-Song Yue, Ph.D

Academic Editor

PLOS ONE

---

## [Editor Report · Acceptance letter]

31 Mar 2021

PONE-D-20-36473R1 

Aquatic invasive alien rodents in western France: where do we stand today after decades of control? 

Dear Dr. Pays:

I'm pleased to inform you that your manuscript has been deemed suitable for publication in PLOS ONE. Congratulations! Your manuscript is now with our production department. 

Kind regards, 

on behalf of

Dr. Bi-Song Yue 

Academic Editor

PLOS ONE